# ContextFormer: Stitching via Latent Expert Calibration

## Abstract

Offline reinforcement learning (RL) algorithms can learn better decision-making compared to behavior policies by stitching the suboptimal trajectories to derive more optimal ones. Meanwhile, Decision Transformer (DT) abstracts the RL as sequence modeling, showcasing competitive performance on offline RL benchmarks. However, recent studies demonstrate that DT lacks of stitching capacity, thus exploiting stitching capability for DT is vital to further improve its performance. In order to endow stitching capability to DT, we abstract trajectory stitching as expert matching and introduce our approach, ContextFormer, which integrates contextual information-based imitation learning (IL) and sequence modeling to stitch sub-optimal trajectory fragments by emulating the representations of a limited number of expert trajectories. To validate our approach, we conduct experiments from two perspectives: 1) We conduct extensive experiments on D4RL benchmarks under the settings of IL, and experimental results demonstrate ContextFormer can achieve competitive performance in multiple IL settings. 2) More importantly, we conduct a comparison of ContextFormer with various competitive DT variants using identical training datasets. The experimental results unveiled ContextFormer's superiority, as it outperformed all other variants, showcasing its remarkable performance.

## 1 Introduction

Depending on whether direct interaction with an environment for acquiring new training samples, reinforcement learning (RL) can be categorized into offline RL (Kumar et al., 2020; Kostrikov et al., 2021) and online RL Haarnoja et al. (2018); Schulman et al. (2017). Among that, offline RL aims to learn the optimal policy from a set of static trajectories collected by behavior policies without the necessity to interact with the online environment Levine et al. (2020). One notable advantage of offline RL is its capacity to learn a more optimal behavior from a dataset consisting solely of sub-optimal trials Levine et al. (2020). This feature renders it an efficient approach for applications where data acquisition is prohibitively expensive or poses potential risks, such as with autonomous vehicles and pipelines. The success of these offline algorithms is attributed to its stitching capability to fully leverage sub-optimal trails and seamlessly stitch them into an optimal trajectory, which has been discussed by Fu et al. (2019; 2020) Different from the majority of offline RL algorithms, Decision Transformer (DT) Chen et al. (2021) abstracts the offline RL problems as a sequence modeling process. Such paradigm achieved commendable performance across various offline benchmarks, including d4rl Fu et al. (2021). Despite its success, recent studies suggest a limitation in DT concerning a crucial aspect of offline RL agents, namely, stitching Yamagata et al. (2023). Specifically, DT appears to fall short in achieving the ability to construct an optimal policy by stitching together sub-optimal trajectories. Consequently, DT inherently lacks the capability to obtain the optimal policy through the stitching of sub-optimal trials. To address this limitation, investigating and enhancing the stitching capability of DT, or introducing additional stitching capabilities, holds the theoretical promise of further elevating its performance in offline tasks.

To endow the stitching capability to the Transformer for decision making, QDT Yamagata et al. (2023) utilizes Q-networks to relabel Return-to-Go (RTG), endowing the stitching capability to DT. While experimental results suggest that relabeling the RTG through a pre-trained conservative Q-network can enhance DT's performance, this relabeling approach with a conservative critic tends to make the learned policy excessively

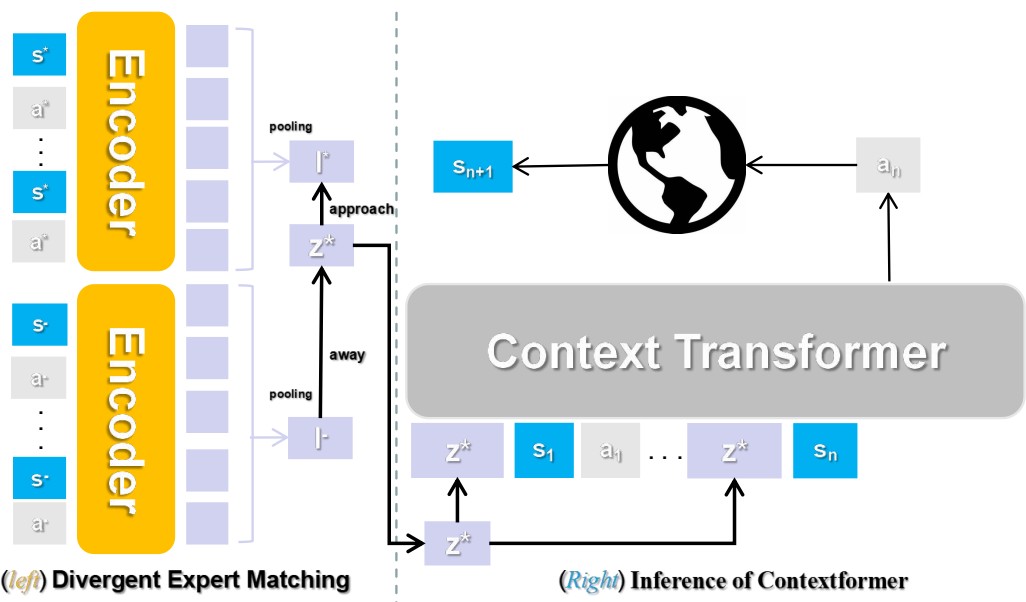

Figure 1: Demonstration of Context Transformer (ContextFormer). (*Right*) ContextFormer utilizes the contextual information $\mathbf{z}^*$ derived from the divergence-based sequential expert matching process to enhance its inference capabilities with the environment. (*left*) $\mathbf{z}^*$ should aim to converge towards the expert contextual information ($I^*$) while distancing itself from sub-optimal contextual information ($I^-$). Consequently, ContextFormer can effectively integrate and stitch together fragments of the in-expert distribution to produce more robust expert-level inferences. **Note**: For a more comprehensive understanding of trajectory stitching, kindly refer to the section **Analysis of Stitching**.

conservative while being suffered from out-of-distribution (OOD) issues. Consequently, the policy's ability to generalize is diminished. To further address this limitation, we approach it from the perspective of supervised and latent (representation)-based imitation learning (IL), *i.e.* expert matching, and propose **ContextFormer**. Specifically, we utilize the representations of a limited number of expert trajectories as demonstrations to stitch sub-optimal trajectories in the latent space. This approach involves the joint and supervised training of a latent-conditioned sequential policy (transformer) while optimizing contextual embedding. By stitching trajectory fragments in the latent space using a supervised training objective, ContextFormer eliminates the need for conservative penalization. Therefore, ContextFormer serves as a remedy for common issues found in both return-conditioned DT and QDT.

To summarize, the majority contribution of our studies can be summarized as follows:

- **We propose a novel IL framework that can endow stitching capability to the transformer for decision making with both theoretical analysis and numerical support.** Specifically, on the theoretical aspect, we demonstrate that expert matching can extract valuable in-expert distributed HI from sub-optimal trajectories, supplying the expert contextual information stitching the in-expert distribution fragments together. On the experimental aspect, extensive experimental results showcase ContextFormer surpass multiple DT variants.

- **Our stitching method is a supervised method, thus getting rid of the limitations of conservatism inherit from offline RL algorithms.** Specifically, our method represents a departure from conservative approaches by adopting a fully supervised objective to enhance the stitching capacity of the Transformer policy. This approach aren't suffered from the OOD samples and the conservatism inherited from the conservative off-policy algorithms.

- **Our approach can extract in-expert distribution information from sub-optimal trails to supplement the contextual embedding which is more informative than Prompt-DT and**

**GDT.** Consequently, ContextFormer can utilize HI from entire dataset for better decision making than Prompt-DT and GDT. Additionally, our method overcomes the constraints posed by scalar reward functions, mitigating information bottlenecks.

## 2 Related Work

**Offline Reinforcement Learning (RL).** Offline RL learns policy from a static offline dataset and lacks the capability to interact with environment to collect new samples for training. Therefore, compared to online RL, offline RL is more susceptible to out-of-distribution (OOD) issues. Furthermore, OOD issues in offline RL have been extensively discussed. The majority competitive methods includes adding regularized terms to the objective function of offline RL to learn a conservative policy Peng et al. (2019); Wu et al. (2022); Chen et al. (2022) or a conservative value network Kumar et al. (2020); Kostrikov et al. (2021); An et al. (2021). By employing such methods, offline algorithms can effectively reduce the overestimation of OOD state actions. Meanwhile, despite the existence of OOD issues in offline RL, its advantage lies in fully utilizing sub-optimal offline datasets to stitch offline trajectory fragments and obtain a better policy Fu et al. (2019; 2021). The ability to enhance the offline learned policy beyond the behavior policy by integrating sub-optimal trajectory fragments is referred to as policy improvement. However, previous researches indicate that DT lacks of stitching capability Therefore, endowing stitching capability to DT could potentially enhance its sample efficiency in offline problem setting. Meanwhile, in the context of offline DT, the baseline most relevant to our study is Q-learning DT Yamagata et al. (2023) (QDT). Specifically, QDT proposes a method that utilizes a conservative critic network trained offline to relabel the RTG in the offline dataset, approximating the capability to stitch trajectories for DT. Unlike QDT, we endow stitching capabilities to DT from the perspective of expert matching that is a supervised and latent-based training objective.

**Imitation Learning (IL).** Previous researches have extensively discussed various IL problem settings and mainly includes LfD Argall et al. (2009); Judah et al. (2014); Ho and Ermon (2016); Brown et al. (2020); Ravichandar et al. (2020); Boborzi et al. (2022), LfO Ross et al. (2011); Liu et al. (2018); Torabi et al. (2019); Boborzi et al. (2022), offline IL Chang et al. (2021); DeMoss et al. (2023); Zhang et al. (2023) and online IL Ross et al. (2011); Brantley et al. (2020); Sasaki and Yamashina (2021). The most related IL methods to our studies are Hindsight Information Matching (HIM) based methods Furuta et al. (2022); Paster et al. (2022); Kang et al. (2023); Liu et al. (2023); Gu et al. (2023), in particular, CEIL Liu et al. (2023) is the novel expert matching approach considering abstract various IL problem setting as a generalized and supervised HIM problem setting. Although both ContextFormer and CEIL share a commonality in calibrating the expert performance via expert matching, different from CEIL that our study focuses on endowing the stitching capabilities to transformer, we additionally utilize sub-optimal trajectory representation to supply the contextual information. Besides, the core contribution of our study is distinct to offline IL that we don't aim to enhance the IL domain but rather to endow stitching capability to transformer for decision making.

## 3 Preliminary

Before formally introducing our framework, we first introduce the basic concepts, which include RL, IL, HIM, and In-Context Learning (ICL).

**Reinforcement Learning (RL).** We consider the sequential decision making process can be represented by a Non-Markov Decision Processing (MDP) tuple, *i.e.* $\mathcal{M} := \big( \mathcal{S}, \mathcal{A}, \mathcal{R}, d_{\mathcal{M}}(\mathbf{s}_{t+1}|\mathbf{s}_t, \mathbf{a}_t), r, \gamma, p(\mathbf{s}_0) \big)$, where $\mathcal{S}$ denotes observation space, $\mathcal{A}$ denotes action space, and $d_{\mathcal{M}}(\mathbf{s}_{t+1}|\mathbf{s}_t, \mathbf{a}_t) : \mathcal{S} \times \mathcal{A} \to \mathcal{S}$ denotes the transition (dynamics) probability, $r(\mathbf{s}_t, \mathbf{a}_t) : \mathcal{S} \times \mathcal{A} \to \mathbb{R}$ denotes the reward function, $\gamma \in [0, 1]$ denotes the discount factor, and $\mathbf{s}_0 \sim p(\mathbf{s}_0)$ is the initial observation, $p(\mathbf{s}_0)$ is the initial state distribution. The goal of sequential decision making is to find the optimal sequence model (policy) termed $\pi^*(\cdot|\tau) : T \times \mathcal{S} \times \mathcal{A} \to \mathcal{A}$ that can bring the highest accumulated return $R(\tau) = \sum_{t=0}^{t=T} {}_{(\mathbf{s}_t,\mathbf{a}_t) \sim \pi} \gamma^t \cdot r(\mathbf{s}_t, \mathbf{a}_t)$, *i.e.* $\pi^* := \arg\max_\pi R(\tau)|_{\tau \sim \pi}$, where $\tau \sim \pi := \big\{ \mathbf{s}_0, \mathbf{a}_0, r(\mathbf{s}_0, \mathbf{a}_0), \cdots, \mathbf{s}_t, \mathbf{a}_t, r(\mathbf{s}_t, \mathbf{a}_t) \big\}$ is the rollout trajectory. Furthermore, DT abstracts offline RL as sequence modeling *i.e.* $\mathbf{a}_t := \pi(\cdot|\hat{R}_0, \mathbf{s}_0, \mathbf{a}_0, \cdots, \hat{R}_t, \mathbf{s}_t)$, where $\hat{R}_{t'} = \sum_{t=t'}^{t=T} \gamma^{t-t'} r(\mathbf{s}_t, \mathbf{a}_t)$ denotes Return-to-Go (RTG).

**Imitation Learning (IL).** In the IL problem setting, the reward function $r(\mathbf{s}_t, \mathbf{a}_t)$ can not be accessed. However, the demonstrations $\mathcal{D}_{\text{demo}} = \left\{ \tau_{\text{demo}} = \{\mathbf{s}_0, \mathbf{a}_0, \cdots, \mathbf{s}_t, \mathbf{a}_t\} | \tau_{\text{demo}} \sim \pi^* \right\}$ or observations $\mathcal{D}_{\text{obs}} = \left\{ \tau_{\text{obs}} = \{\mathbf{s}_0, \cdots, \mathbf{s}_t\} | \tau_{\text{obs}} \sim \hat{\pi} \right\}$ are available. Accordingly, the goal of IL is to recover the performance of expert policy by utilizing extensive sub-optimal trajectories $\hat{\tau} \sim \hat{\pi}$ imitating expert demonstrations or observations, where $\hat{\pi}$ is the sub-optimal policy. Meanwhile, according to the objective of IL, it has two general settings: 1) In the setting of LfD, we imitate from demonstration. 2) In the setting of LfO, we imitate from observation.

**Hindsight Information Matching (HIM).** Furuta et al. define the HIM as training conditioned policy with HI *i.e.* learning a contextual policy $\pi(\cdot | \mathbf{z}, \mathbf{s}) : \mathcal{Z} \times \mathcal{S} \to \mathcal{A}$ by Equation 1.

$$\pi(\cdot | \mathbf{z}, \mathbf{s}) := \arg\min_\pi \mathbb{E}_{\mathbf{z} \sim p(\mathbf{z}), \tau_\mathbf{z} \sim \pi_\mathbf{z}}[D(\mathbf{z}, I_\phi(\tau_\mathbf{z}))], \tag{1}$$

where $I_\phi(\tau_\mathbf{z}) : \mathcal{S} \times \mathcal{A} \to \mathcal{Z}$ denotes the statistical function that can extract representation or HI from the $\mathbf{z}$ conditioned offline trajectory $\tau_\mathbf{z}$ *i.e.* $\mathbf{z}_\tau = I_\phi(\tau_\mathbf{z})$ (when utilizing $\tau_\mathbf{z}$ to compute $\mathbf{z}_\tau$, we remove $\mathbf{z}$ from $\tau_\mathbf{z}$) and $D$ denotes the metric used to estimate the divergence between the initialized latent representation $\mathbf{z} \sim p(\mathbf{z})$ and the trajectory HI *i.e.* $I_\phi(\tau_\mathbf{z})$, where $\pi_\mathbf{z} := \{\mathbf{z}, \mathbf{s}_0, \mathbf{a}_0, \cdots, \mathbf{z}, \mathbf{s}_t, \mathbf{a}_t\}$. In particular, $\tau_\mathbf{z}$ will be optimal once we set up $\mathbf{z}$ as $\mathbf{z} := \arg\min_\mathbf{z} D(\mathbf{z}, I_\phi(\tau^*)) |_{\tau^* \sim \pi^*(\tau)}$.

**In-Context Learning (ICL).** Xu et al. showcases that DT can be prompted with offline trajectory fragments to conduct fine-tuning and adaptation on new similar tasks. *i.e.* $\mathbf{a}_t := \pi(\cdot | \tau_{\text{prompt}} \oplus \{\mathbf{s}_0, \mathbf{a}_0, \cdots, \mathbf{s}_t\})$, where $\oplus$ denotes concatenation, and the prompt is: $\tau_{\text{prompt}} = \{\hat{\mathbf{s}}_0, \hat{\mathbf{a}}_0, \cdots, \hat{\mathbf{s}}_k, \hat{\mathbf{a}}_k\}$. Despite that ContextFormer also utilizes the contextual information. However, it is different from Prompt-DT that ContextFormer condition on the offline trajectory's latent representation rather prompt to inference, which enables the long horizontal information being embedded into the contextual embedding, and such contextual embedding contains richer information than prompt-based algorithm, therefore, contextual embedding is possible to consider useful samples with longer-term future information during the training process.

## 4 Can expert matching endow stitching to transformer for decision making?

DT lacks of stitching capacity has been noted in previous studies. Consequently, it is imperative to investigate methods to augment this capability and enhance the overall performance of DT. One such proposed solution is Q-learning DT Yamagata et al. (2023), which involves leveraging a pre-trained conservative Q network to relabel the Return-to-Go (RTG) values of offline RL datasets. Subsequently, DT undergoes training on the relabeled dataset to acquire stitching capacity. However, this approach has several limitations. During the evaluation process, DT may encounter out-of-distribution samples, potentially disrupting its decision-making process. In contrast to Q-DT, ContextFormer utilizes divergent sequential expert matching (definition 2) to endow DT with stitching capacity. In particular, different from previous expert matching approach Liu et al. (2023), which solely mimic the expert policy for decision-making, divergent sequential expert matching goes a step further by harnessing in-expert distribution HI from sub-optimal datasets (Theorem 5.1). By seamlessly stitching them together, it eliminates the overestimation of scarcity in expert demonstrations.

In the upcoming sections, we will elucidate how divergent sequential expert matching extracts in-expert HI from sub-optimal datasets and adeptly integrates them.

**Notations.** In this section we provide the notations we utilized. Specifically, we define the expert policy (sequential policy) as $\pi^*(\cdot | \tau^*)$, the sub-optimal policy as $\hat{\pi}(\cdot | \hat{\tau})$, and the density functions of expert and sub-optimal policies are respectively represented as $P^*(\tau) \sim \pi^*$, $\hat{P}(\tau) \sim \hat{\pi}$. Furthermore, we define the optimal (expert) trajectory as $\tau^* \sim \pi^*(\tau)$, the sub-optimal trajectory as $\hat{\tau} \sim \hat{\pi}(\tau)$, and the mixture of expert and sub-optimal trajectories as $\tau \sim \pi^*(\tau)$ and $\hat{\pi}(\tau)$. Subsequently, we define latent conditioned sequence modeling as Definition 1, expert matching based IL as Definition 2.

**Definition 1** (Latent conditioned sequence modeling)**.** *Given the latent embedding $\mathbf{z}$, the process of latent conditioned sequence modeling can be formulated as $\mathbf{a}_t := \pi_\mathbf{z}(\cdot | \mathbf{z}, \mathbf{s}_0, \mathbf{a}_0, \cdots, \mathbf{z}, \mathbf{s}_t)$, where $D(\mathbf{z} || I_\phi(\tau_\mathbf{z})) \le \epsilon$, $\epsilon$*

*is a very small threshold, $D$ is divergence function, $t \in [0,T]$ Meanwhile, we define $\pi_{I_\phi(\tau)}$ as $\pi_{\mathbf{z}}$ conditioned on $I_\phi(\tau)$*

Previously, Liu et al. propose utilizing $\mathbf{z}^*$ solely to mimic the expert HI. However, if the expert demonstrations are not sufficient to estimate a robust representation, it may limit the generality of the contextual policy. In order to further enhance the estimation of $\mathbf{z}^*$, we propose divergent sequential expert matching as defined in Definition 2.

**Definition 2** (Divergent Sequential Expert Matching). *Given the HI extractor $I_\phi(\cdot|\tau)$, the process of divergent sequential expert matching can be defined as: jointly optimizing the contextual information (or HI) $\mathbf{z}^*$ and contextual policy $\pi_{\mathbf{z}}(\cdot|\tau)$ to robustly match the expert policy i.e. $\pi_{\mathbf{z}^*} := \arg\min_{\pi_{\mathbf{z}^*}} D(I_\phi(\tau^*)||\mathbf{z}^*) - D(I_\phi(\hat{\tau})||\mathbf{z}^*) + D\big(\pi_{I_\phi(\tau)}(\cdot|\tau_{I_\phi(\tau)})||\pi(\cdot|\tau)\big)$.*

As mentioned in Definition 2, the optimal contextual embedding $\mathbf{z}^*$ should have to be calibrated with the the expert trajectory's HI and away from the sub-optimal trajectory's HI. Furthermore, based on Definition 2, we propose a more concise expression *i.e.*

$$\min \mathcal{J}(\mathbf{z}^*) = \min_{\mathbf{z}^*, I_\phi} \mathbb{E}_{\tau^* \sim \pi^*(\tau)}[\lambda_1 \cdot ||\mathbf{z}^* - I_\phi(\tau^*)||] - \mathbb{E}_{\hat{\tau} \sim \hat{\pi}}[\lambda_2 \cdot ||\mathbf{z}^* - I_\phi(\hat{\tau})||], \tag{2}$$

where $\lambda_1$, $\lambda_2$ separately denote the weight. Subsequently, we analyze why Equation 2 can stitch sub-optimal fragments.

## 5 Analysis of stitching

We regard $P^*(\tau)$ or $\hat{P}(\tau)$ as density function, separately estimating the probability of $\tau$ being sampled from policies $\pi^*(\cdot|\tau^*)$ and $\hat{\pi}(\cdot|\hat{\tau})$. Subsequently, we propose Theorem 5.1:

**Theorem 5.1** (Expert Calibration). *Given the expert policy $\pi^*(\cdot|\tau)$, the sub-optimal policy $\hat{\pi}(\cdot|\tau)$, the HI extractor $I_\phi(\cdot|\tau)$, contextual embedding $\mathbf{z}^*$. Minimizing Equation 2 is equivalent to:*

$$\min_{\mathbf{z}^*, I_\phi} K \cdot \Bigg( \underbrace{\int_{\tau \sim \mathcal{S} \times \mathcal{A}} \mathbb{1}(\lambda_1 \cdot P^*(\tau) \geq \lambda_2 \cdot \hat{P}(\tau))||\mathbf{z}^* - I_\phi(\tau)||d\tau}_{J_{\text{term1}}}$$

$$+ \underbrace{\int_{\tau \sim \mathcal{S} \times \mathcal{A}} \mathbb{1}(\lambda_1 \cdot P^*(\tau) \leq \lambda_2 \cdot \hat{P}(\tau))||\mathbf{z}^* - I_\phi(\tau)||d\tau}_{J_{\text{term2}}} \Bigg)$$

*, where $\mathbb{1}$ denotes indicator, and $K = (\lambda_1 P^*(\tau) - \lambda_2 \hat{P}(\tau))$. Proof of Theorem 5.1 see Appendix.*

**Connection with Stitching.** It can be concluded from Theorem 5.1 that when $\lambda_1 \cdot P^*(\tau) \geq \lambda_2 \cdot \hat{P}(\tau)$ *i.e.* current trajectory fragments are much more possible sampled from expert policy, $\mathbf{z}^*$ will be away from its hindsight information, vice visa. Therefore, Theorem 5.1 demonstrates the capability of divergent sequential expert matching to leverage $\mathbf{z}^*$ to extract HI from trajectory fragments aligning with expert HI within trajectories (we provide a case in Figure 2). Consequently, $\pi_{\mathbf{z}}$ can generate trajectories conforming to the in-expert distribution when conditioned on $\mathbf{z}^*$, and stitch the in-expert distribution fragments together. In this following section, we propose our approach ContextFormer.

## 6 Context Transformer (ContextFormer)

We introduce ContextFormer, which utilizes divergent sequential expert matching to empower latent conditioned Transformer with stitching capabilities, stitching the in-expert trajectory fragments in latent space. Furthermore, compared to previous scalar-conditioned DT and its variants, ContextFormer is conditioned on more informative factors, thereby overcoming the limitations of information bottlenecks. Meanwhile, ContextFormer's objective is entirely supervised, thus overcoming the conservatism limitations of DT variants inherent in jointly utilizing offline RL algorithms.

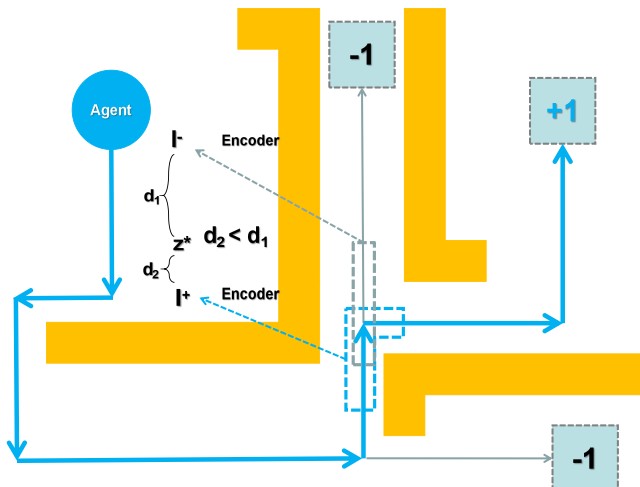

Figure 2: Demonstration of Stitching. $I^*$ represents the HI that is associated with being close to an expert trajectory, while $I^-$ represents the HI that is associated with being far from an expert trajectory.

## 6.1 Method

**Training Procedure.** We model the contextual sequence model by the aforementioned latent conditioned sequence modeling as defined in Definition 1, and the supervised policy loss defined as Equation 3. Meanwhile, we optimize $\mathbf{z}^*$ by training the contextual sequence model with the expert matching objective as defined in Definition 2

$$\mathcal{J}_{\pi_\mathbf{z}, I_\phi} = \mathbb{E}_{\tau \sim (\pi^*, \hat{\pi})}[||\pi(\cdot|I_\phi(\tau), \mathbf{s}_0, \mathbf{a}_0, \cdots, I_\phi(\tau), \mathbf{s}_t) - \mathbf{a}_t||], \tag{3}$$

where $\tau = \{\mathbf{s}_0, \mathbf{a}_0, \cdots, \mathbf{s}_t, \mathbf{a}_t\}$ is the rollout trajectory, while $\hat{\pi}$ and $\pi^*$ are separated to the sub-optimal and optimal policies, and $\pi_{I_\phi(\tau)}(\cdot|\tau_{I_\phi}) = \pi(\cdot|I_\phi(\tau_\mathbf{z}^{t-k:t}), \mathbf{s}_0, \mathbf{a}_0, \cdots, I_\phi(\tau_\mathbf{z}^{t-k:t}), \mathbf{s}_t)$. Meanwhile, we also optimize the HI extractor $I_\phi$ and contextual embedding $\mathbf{z}^*$ via Equations 3 and 4:

$$\mathcal{J}_{\mathbf{z}^*, I_\phi} = \min_{\mathbf{z}^*, I_\phi} \mathbb{E}_{\hat{\tau} \sim \hat{\pi}(\tau), \tau^* \sim \pi^*(\tau)}[||\mathbf{z}^* - I_\phi(\tau^*)|| - ||\mathbf{z}^* - I_\phi(\hat{\tau})||] \tag{4}$$

**Evaluation Procedure.** Based on the modeling approach defined in Definition 1, we utilize the contextual embedding $\mathbf{z}^*$ optimized by Equation 2 as the goal for each inference moment of our latent conditioned sequence model, thereby auto-regressively rolling out trajectory in the environment to complete the testing (Algorithm 1).

## 6.2 Practical Implementation of ContextFormer

We utilize BERT Devlin et al. (2019) as the defined HI extractor $I_\phi$, and randomly initialize a vector as the contextual embedding $\mathbf{z}^*$. Our contextual policy $\pi_\mathbf{z}(\cdot|\tau_\mathbf{z}^{t-k:t})$ is modified from the DT Chen et al. (2021) that we replace $\hat{R}$ with $\mathbf{z}^*$. The input of ContextFormer is a trajectory fragments with a window size of k *i.e.* $\tau_\mathbf{z}^{t-k:t} = \{\mathbf{z}, \mathbf{s}_{t-k}, \mathbf{z}, \mathbf{a}_{t-k}, \cdots, \mathbf{z}, \mathbf{s}_t\}$. For more detials about ContextFormer's hyperparameter, please refer to Experimental Setup section of Appendix. In terms of our training framework, as shown in Algorithm 1, the optimization of $I_\phi$ involves the joint utilization of the divergent sequential expert matching objective as formulated in Equation 2 and the latent conditioned supervised training loss as formulated in Equation 3. Notably, Equation 2 and Equation 3 are not used simultaneously to optimize $\mathbf{z}^*$. Instead, in each updating epoch, we use Equation 3 to update $I_\phi$ and $\pi_\mathbf{z}$. Subsequently, we freeze $\pi_\mathbf{z}$ and then use Equation 2 to update $I_\phi$. Finally, with $I_\phi$ and $\pi_\mathbf{z}$ frozen, we use Equation 2 to update $\mathbf{z}^*$. The evaluation process has also been depicted in Algorithm 1 that once we obtain $\mathbf{z}^*$ and $\pi_\mathbf{z}$, we autoregressively utilize latent conditioned sequence modeling to conduct evaluation.

---

**Algorithm 1** ContextFormer

---

**Require:** HIM extractor $I_\phi(\cdot|\tau)$, Contextual policy $\pi_\mathbf{z}(\cdot|\tau)$, sub-optimal offline datasets $D_{\hat\tau} \sim \hat\pi$, randomly initialized contextual embedding $\mathbf{z}^*$, and demonstrations (expert trajectories) $D_{\tau^*} \sim \pi^*$

**Training:**

1: Sample batch suboptimal trails $\hat\tau$ from $\mathcal{D}_{\hat\tau}$, and sampling batch demonstrations $\tau^*$ from $\mathcal{D}_{\pi^*}$.
2: Update HI extractor $I_\phi$ by solving Equation 3, Equation 4. Update $\mathbf{z}^*$ by solving Equation 4.
3: Update policy $\pi_\mathbf{z}$ by solving Equation 3.

**Evaluation:**

1: Initialize $t = 0$; $\mathbf{s}_t \leftarrow$ env.reset(); $\tau = \{\mathbf{s}_0\}$; done = False, $R = 0$, $N = 0$.
2: **while** $t \leq N$ or not done **do**
3:    $\mathbf{a}_t \leftarrow \pi(\cdot|\tau_t)$;
4:    $\mathbf{s}_{t+1}, \text{done}, r_t \leftarrow$ env.step($\mathbf{a}_t$);
5:    $\tau$.append($\mathbf{a}_t, \mathbf{s}_{t+1}$)
6:    $R+ = r_t; t+ = 1$
7: **end while**
8: Return $R$

---

## 6.3 Experimental settings

Table 1: Normalized scores (averaged over 10 trails for each task) when we vary the number of the expert demonstrations (#5, #20). The highest scores are highlighted.

| | Offline IL Algorithm | Hopper | | | Halfcheetah | | | Walker2d | | | Ant | | | sum |
|---|---|---|---|---|---|---|---|---|---|---|---|---|---|---|
| | | m | mr | me | m | mr | me | m | mr | me | m | mr | me | |
| LfD #5 | ORIL (TD3+BC) | 42.1 | 26.7 | 51.2 | 45.1 | 2.7 | 79.6 | 44.1 | 22.9 | 38.3 | 25.6 | 24.5 | 6.0 | 408.8 |
| | SQIL (TD3+BC) | 45.2 | 27.4 | 5.9 | 14.5 | 15.7 | 11.8 | 12.2 | 7.2 | 13.6 | 20.6 | 23.6 | -5.7 | 192.0 |
| | IQ-Learn | 17.2 | 15.4 | 21.7 | 6.4 | 4.8 | 6.2 | 13.1 | 10.6 | 5.1 | 22.8 | 27.2 | 18.7 | 169.2 |
| | ValueDICE | 59.8 | 80.1 | 72.6 | 2.0 | 0.9 | 1.2 | 2.8 | 0.0 | 7.4 | 27.3 | 32.7 | 30.2 | 316.9 |
| | DemoDICE | 50.2 | 26.5 | 63.7 | 41.9 | 38.7 | 59.5 | 66.3 | 38.8 | 101.6 | 82.8 | 68.8 | 112.4 | 751.2 |
| | SMODICE | 54.1 | 34.9 | 64.7 | 42.6 | 38.4 | 63.8 | 62.2 | 40.6 | 55.4 | 86.0 | 69.7 | 112.4 | 724.7 |
| | CEIL | 94.5 | 45.1 | 80.8 | 45.1 | 43.3 | 33.9 | 103.1 | 81.1 | 99.4 | 99.8 | 101.4 | 85.0 | 912.5 |
| | ContextFormer | 74.9 | 77.8 | 103.0 | 43.1 | 39.6 | 46.6 | 80.9 | 78.6 | 102.7 | 103.1 | 91.5 | 123.8 | **965.6** |
| LfO #20 | ORIL (TD3+BC) | 55.5 | 18.2 | 55.5 | 40.6 | 2.9 | 73.0 | 26.9 | 19.4 | 22.7 | 11.2 | 21.3 | 10.8 | 358.0 |
| | SMODICE | 53.7 | 18.3 | 64.2 | 42.6 | 38.0 | 63.0 | 68.9 | 37.5 | 60.7 | 87.5 | 75.1 | 115.0 | 724.4 |
| | CEIL | 44.7 | 44.2 | 48.2 | 42.4 | 36.5 | 46.9 | 76.2 | 31.7 | 77.0 | 95.9 | 71.0 | 112.7 | 727.3 |
| | ContextFormer | 67.9 | 77.4 | 97.1 | 43.1 | 38.8 | 55.4 | 79.8 | 79.9 | 109.4 | 102.4 | 86.7 | 132.2 | **970.1** |

**Imitation Leaning (IL).** These IL experiment aims to validate our claim (Section 5) that divergent sequential expert matching can extract in-expert HI from sub-optimal trails to provide $\mathbf{z}^*$. Intuitively, the better performance achieved in these tasks, the stronger the validation of our claim. We utilize 5 to 20 expert trajectories and conduct evaluations under both LfO and LfD settings. The objective of these tasks is to emulate $\pi^*(\cdot|\tau)$ by leveraging a substantial amount of sub-optimal offline dataset $\hat\tau \sim \hat\pi$, aiming to achieve performance that equals or even surpasses that of the expert policy $\pi^*(\cdot|\tau)$. In particular, when conduct LfO setting, we imitate from $\tau_{\text{obs}}$, when conduct LfD setting we imitate from $\tau_{\text{demo}}$ (both $\tau_{\text{demo}}$ and $\tau_{\text{obs}}$ are mentioned in section **Preliminary**). And we utilize $\tau_{\text{demo}}/\tau_{\text{obs}}$ as $\tau^*$ (contextual optimization) in LfD and LfO settings to optimize $\mathbf{z}^*$ by Equation 4 and Equation 3. We optimize $\pi_\mathbf{z}(\cdot|\tau_\mathbf{z})$ by Equation 3.

**DT comparisons.** These experiments are conducted to substantiate our contributions, as evidenced by the expectation that ContextFormer should outperform various selected DT baselines. Note that, we are disregarding variations in experimental settings such as IL, RL, etc. Our focus is on controlling factors (data composition, model architecture, etc.), with a specific emphasis on comparing model performance. In particular, ContextFormer is trained under the same settings as **IL**, while DT variants underwent evaluation using the original settings.

### 6.4 Training datasets

**IL.** Our experiments are conducted on four Gym-Mujoco Brockman et al. (2016) environments, including Hopper-v2, Walker2d-v2, Ant-v2, and HalfCheetah-v2. These tasks are constructed utilizing D4RL Fu et al. (2021) datasets including `medium-replay` (mr), `medium` (m), `medium-expert` (me), and `expert` (exp).

**DT comparisons.** When comparing ContextFormer with DT, GDT, and Prompt-DT, we utilize the datasets discussed in **IL**. Additionally, we compare QDT and ContextFormer on the maze2d domain, specifically designed to assess their stitching abilities Yamagata et al. (2023). In terms of the dataset we utilize for training baselines, we ensure consistency in our comparisons by using identical datasets. For instance, when comparing ContextFormer (LfD #5) on Ant-medium, we train DT variants with the same datasets (5 expert trails+ all medium trails).

### 6.5 Baselines

**Imitation learning.** Our IL baselines include ORIL Zolna et al. (2020), SQIL Reddy et al. (2019), IQ-Learn Garg et al. (2022), ValueDICE Kostrikov et al. (2019), DemoDICE Kim et al. (2022), SMODICE Ma et al. (2022), and CEIL Liu et al. (2023). The results of these baselines are directly referenced from Liu et al. (2023), which are utilized to be compared with ContextFormer in both the LfO and LfD settings, intuitively showcasing the transformer's capability to better leverage sub-optimal trajectories with the assistance of expert trajectories (hindsight information). **DT comparisons.** To further demonstrate the superiority of ContextFormer, we carry out comparisons between ContextFormer and DT, DT variants (Prompt-DT, PTDT-offline, and QDT), utilizing the same dataset. This involves comparing the training performance of various transformers.

### 6.6 Results of IL experiments

**ContextFormer demonstrates competitive performance in leveraging expert information to learn from sub-optimal datasets.** We conduct various IL task settings including LfO and LfD to assess the performance of ContextFormer. As illustrated in table 6.6, ContextFormer outperforms selected baselines, achieving the highest performance in both LfD #5 and LfO #20 settings, showcasing respective improvements of 5.8% and 33.4% compared to the best baselines (CEIL). Additionally, ContextFormer closely approach CEIL under the LfO #20 setting. These results demonstrate the effectiveness of our approach in utilizing expert information to assist in learning the sub-optimal dataset, which is cooperated with our analysis (Section 5).

Table 2: Comparison of performance between DT and ContextFormer (LfD #5).

| Task | DT+5 *exp traj* | DT+10 *exp traj* | ContextFormer (LfD #5) |
|---|---|---|---|
| `hopper-m` | 69.5±2.3 | 72.0±2.6 | **74.9**±9.5 |
| `walker2d-m` | 75.0±0.7 | 75.7±0.4 | **80.9**±1.3 |
| `halfcheetah-m` | 42.5±0.1 | 42.6±0.1 | **43.1**±0.2 |
| `hopper-mr` | 78.9±4.7 | **82.2**±0.5 | 77.8±13.3 |
| `walker2d-mr` | 74.9±0.3 | 78.3±5.6 | **78.6**±4.0 |
| `halfcheetah-mr` | 37.3±0.4 | 37.6±0.8 | **39.6**±0.4 |
| **sum** | 378.1 | 388.4 | **394.9** |

### 6.7 ContextFormer showcase better performance than various DT baselines

**ContextFormer vs. Return Conditioned DT.** ContextFormer leverages contextual information as condition, getting riding of limitations such as the information bottleneck associated with scalar return. Additionally, we highlighted that expert matching aids the Transformer in stitching sub-optimal trajectories.

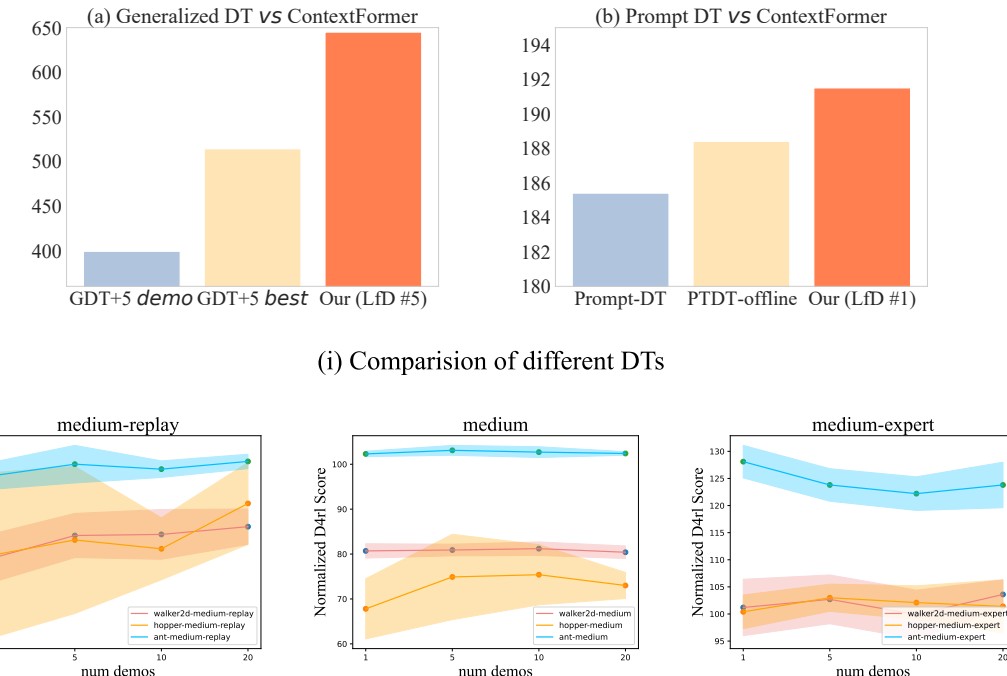

(i) Comparision of different DTs

(ii) Variation in the number of demos

Figure 3: (i) Total Normalized Scores of ContextFormer, GDT and Prompt DT. (i.a) Performance comparison with Generalized DT. (i.b) Performance comparison with Prompt-DT. Specifically, we conducted a performance comparison between ContextFormer (LfD #5) and GDT using the same six offline datasets: `hopper-m (mr)`, `walker2d-m (mr)`, and `halfcheetah-m (mr)`. Additionally, we compared ContextFormer (LfD #1) with Prompt-DT and PTDT-offline on `hopper-m`, `walker2d-m`, and `halfcheetah-m`. The original experimental results have been appended in Appendix. (ii) In this figure, we gradually increase the descriptions of expert trajectories and further observe the performance of ContextFormer in the Learning from Demonstration (LfD) setting.

Therefore, the performance of ContextFormer is expected to be better than DT when using the same training dataset. To conduct this comparison, we tested DT and ContextFormer on `medium-replay`, `medium`, and `medium-expert` offline datasets. As shown in table 2, ContextFormer (LfD #5) demonstrated approximately a 4.4% improvement compared to DT+5 expert trajectories (*exp traj*) and a 1.7% improvement compared to DT+10 *exp traj*.

**ContextFormer vs. GDT and Prompt-DT.** ContextFormer, Prompt-DT, and G-DT all use contextual information for decision-making. However, ContextFormer differs from Prompt-DT and GDT in the way and efficiency of utilizing contextual information. Firstly, The contextual information of ContextFormer fuses in-expert HI extracted from entire datasets. But the Prompt-DT only encompass trajectory fragments, and GDT only encompass local representations. Additionally, according to the insights of HIM, incorporating richer and more diverse future information into the contextual information can help generate more varied trajectories, thereby improving generality. Accordingly, since $\mathbf{z}^*$ encompasses significantly more expert-level HI than isolated trajectory fragments, it becomes easier to generate much more near-expert trajectories by conditioning ContextFormer on $\mathbf{z}^*$. As shown in Figure 3 (a), we conduct a performance comparison between ContextFormer (LfD #5) and GDT (*GDT includes BDT and CDT, we utilized BDT as our baseline*) using the same 6 offline datasets including `hopper-m (mr)`, `walker2d-m (mr)`, `halfcheetah-m (mr)`. ContextFormer demonstrates a remarkable 25.7% improvement compared to the best GDT setting. As shown in Figure 3 (b), we compare ContextFormer (LfD #1) with Prompt-DT and PTDT-offline on `hopper-m`, `walker2d-m`,

`halfcheetah-m`. The experimental results demonstrate that ContextFormer outperforms PTDT-offline by 1.6%, Prompt-DT by 3.3%. Therefore, the ability of ContextFormer to utilize contextual information has been validated.

Table 3: Comparison of the performance difference between QDT and ContextFormer (LfD #10). ContextFormer (LfD #10) performs the best. Notably, the experimental results of QDT, DT and CQL are directly quated from Yamagata et al. (2023).

| Task | QDT | DT | ContextFormer (LfD #top 10 $\tau$) |
|------|-----|-----|------------------------------------|
| `maze2d-open-v0` | 190.1±37.8 | 196.4±39.6 | 204.2±13.3 |
| `maze2d-medium-v1` | 13.3±5.6 | 8.2±4.4 | 63.6±25.6 |
| `maze2d-large-v1` | 31.0±19.8 | 2.3±0.9 | 33.8±12.9 |
| `maze2d-umaze-v1` | 57.3±8.2 | 31.0±21.3 | 61.8±0.1 |
| **sum** | 291.7 | 237.9 | **363.4** |

**ContextFormer vs Q-DT.** QDT utilizes a pre-trained conservative Q network to relabel the offline dataset, thereby endowing stitching capability to DT for decision-making. Our approach differs from QDT in that we leverage representations of expert trajectories to stitch sub-optimal trajectories. The advantages of ContextFormer lie in two aspects. On the one hand, our method can overcome the information bottleneck associated with the scalar reward function. On the other hand, our objective is a supervised objective, thereby eliminating the constraints of a conservative policy. Meanwhile, as demonstrated in Table 3, we evaluate ContextFormer on multiple tasks in the maze2d domain, utilizing the top 10 trials ranked by return as demonstrations. Our algorithm achieves a score of 364.3, surpassing all DTs.

**Impact of the number of demonstrations.** We vary the number of $\tau_{\text{demo}}$ to conduct evaluation. Specifically, as illustrated in Figure 3 (ii), ContextFormer's performance on `medium-replay` tasks generally improves with an increasing number of demonstrations. However, for `medium` and `medium-expert` datasets, there is only a partial improvement trend with an increasing number of $\tau_{\text{demo}}$. In some `medium-expert` tasks, there is even a decreasing trend. This can be attributed to the diverse trajectory fragments in `medium-replay`, enabling ContextFormer to effectively utilize expert information for stitching sub-optimal trajectory fragments, resulting in improved performance on `medium-replay` tasks. However, in `medium` and `medium-expert` tasks, the included trajectory fragments may not be diverse enough, and expert trajectories in the `medium-expert` dataset might not be conducive to effective learning. As a result, ContextFormer exhibits less improvement on `medium` datasets and even a decrease in performance on `medium-expert` tasks.

**Impact of demonstrations' diversity.** Sown in Figure 4 (a). To demonstrate the influence of diversity among demonstrations on ContextFormer's performance, we initially identify a trajectory with the highest return, denoted as the **best traj**. Subsequently, we select 100 trajectories with returns similar to, but varying from, this reference trajectory (with differences in returns falling within the range [0, 100]). Following this, we arrange all trajectories in ascending order based on the cosine similarity of their states with those of the **best traj** in the sorted list. We then sample 10 trajectories using the following strategies: uniformly sampling from the `far left`, `left quarter`, `mid` and `right quarter`. The experimental results suggest a positive correlation between the diversity of demos and the performance of ContextFormer.

**Impact of demonstrations's quality.** As shown in Figure 4 (b), we first arrange all expert trajectories according to their returns. Subsequently, we sample four demonstration sets by shifting a window of 10 steps across various starting points within the sorted queue: `left quarter`, `mid`, and the position ten steps before the `far right`. The experimental demonstrates a trend: higher-quality demonstrations are generally related to higher performance.

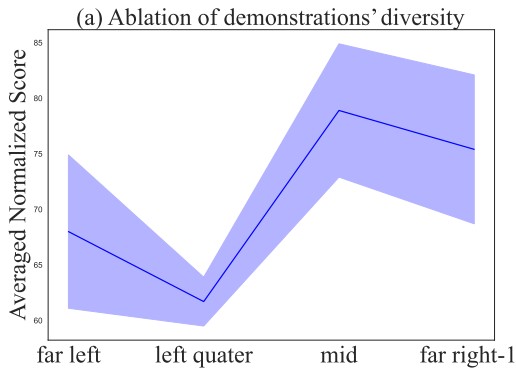
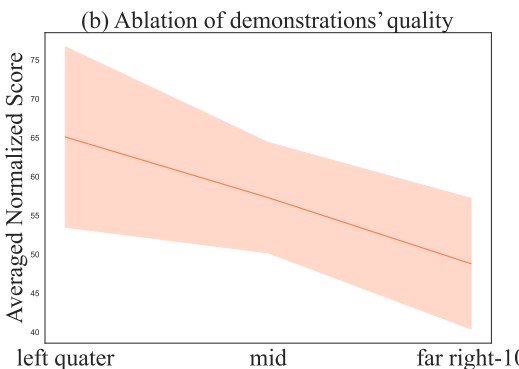

Figure 4: Ablations of demonstration. (a) Impact of demonstrations' diversity. (b) Impact of demonstrations' quality.

## 7 Conclusions

We empower the Transformer with stitching capabilities for decision-making by leveraging expert matching and latent conditioned sequence modeling. Our approach achieves competitive performance on IL tasks, surpassing all selected DT variants on the same dataset, thus demonstrating its feasibility. Furthermore, from a theoretical standpoint, we provide mathematical derivations illustrating that stitching sub-optimal trajectory fragments in the latent space enables the Transformer to infer necessary decision-making aspects that might be missing in sub-optimal trajectories.

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

# A  Mathematics Proof.

## A.1  Proof of Theorem 5.1

Given our contextual optimization objective:

$$\mathcal{J}_{I_\phi, \mathbf{z}^*} = \min_{\mathbf{z}^*, I_\phi} \mathbb{E}_{\hat{\tau} \sim \hat{\pi}_{\mathbf{z}^*}, \tau^* \sim \pi^*} \left[ \lambda_1 \cdot \left|\left| \mathbf{z}^* - I_\phi(\tau^*) \right|\right| - \lambda_2 \cdot \left|\left| \mathbf{z}^* - I_\phi(\hat{\tau}) \right|\right| \right] \tag{5}$$

we regard $P^*(\tau)$ or $\hat{P}(\tau)$ as density function, separately estimating the probability of $\tau$ being sampled from policies $\pi^*(\cdot|\tau^*)$ and $\hat{\pi}(\cdot|\hat{\tau})$.

Then we first introduce importance sampling *i.e.* $\int_{\tau^* \sim \pi^*(\tau)} f(\tau^*) d\tau^* = \int_{\hat{\tau} \sim \hat{\pi}} \frac{P^*(\tau)}{\hat{P}(\tau)} f(\hat{\tau}) d\hat{\tau}$.

And, we introduce: the transformation of sampling process from local domain to global domain *i.e.* $\int_{\tau^* \sim \pi^*} f(\tau^*) = \int_{\tau \sim \mathcal{S} \times \mathcal{A}} P^*(\tau) \cdot f(\tau) d\tau$, where $f(\tau)$ denotes the objective function.

Based on above, we derivative:

$$
\begin{aligned}
\mathcal{J}_{I_\phi, \mathbf{z}^*} &= \min_{\mathbf{z}^*, I_\phi} \mathbb{E}_{\tau^* \sim \pi^*(\tau)}[\lambda_1 \cdot ||\mathbf{z}^* - I_\phi(\tau^*)||] - \mathbb{E}_{\hat{\tau} \sim \hat{\pi}(\tau)}[\lambda_2 \cdot ||\mathbf{z}^* - I_\phi(\hat{\tau})||] \\
&= \min_{\mathbf{z}^*, I_\phi} \mathbb{E}_{\hat{\tau} \sim \hat{P}(\tau)} [\frac{\lambda_1 \cdot P^*(\tau)}{\hat{P}(\tau)} ||\mathbf{z}^* - I_\phi(\hat{\tau})||] - \mathbb{E}_{\hat{\tau} \sim \hat{\pi}(\tau)}[\lambda_2 \cdot ||\mathbf{z}^* - I_\phi(\hat{\tau})||] \\
&= \min_{\mathbf{z}^*, I_\phi} \mathbb{E}_{\hat{\tau} \sim \hat{\pi}(\tau)} [(\frac{\lambda_1 \cdot P^*(\tau)}{\hat{P}(\tau)} - \lambda_2) ||\mathbf{z}^* - I_\phi(\hat{\tau})||] \\
&= \min_{\mathbf{z}^*, I_\phi} \int_{\tau \sim \mathcal{S} \times \mathcal{A}} \hat{P}(\tau) \left( \frac{\lambda_1 \cdot P^*(\tau)}{\hat{P}(\tau)} - \lambda_2 \right) ||\mathbf{z}^* - I_\phi(\tau)|| d\tau \\
&= \min_{\mathbf{z}^*, I_\phi} \int_{\tau \sim \mathcal{S} \times \mathcal{A}} \left( \lambda_1 \cdot P^*(\tau) - \lambda_2 \cdot \hat{P}(\tau) \right) ||\mathbf{z}^* - I_\phi(\tau)|| d\tau \\
&= \min_{\mathbf{z}^*, I_\phi} \underbrace{\int_{\tau \sim \mathcal{S} \times \mathcal{A}} \mathbb{1}(\lambda_1 \cdot P^*(\tau) \geq \lambda_2 \cdot \hat{P}(\tau)) \left( \lambda_1 P^*(\tau) - \lambda_2 \hat{P}(\tau) \right) ||\mathbf{z}^* - I_\phi(\tau)|| d\tau}_{J_{\text{term1}}} \\
&\quad + \underbrace{\int_{\tau \sim \mathcal{S} \times \mathcal{A}} \mathbb{1}(\lambda_1 \cdot P^*(\tau) \leq \lambda_2 \cdot \hat{P}(\tau)) \left( \lambda_1 P^*(\tau) - \lambda_2 \hat{P}(\tau) \right) ||\mathbf{z}^* - I_\phi(\tau)|| d\tau}_{J_{\text{term2}}},
\end{aligned}
\tag{6}
$$

where $\mathbb{1}$ denotes indicator.

# B  Experimental Setup

## B.1  Model Hyperarameters

The hyperparameter settings of our customed Decision Transformer is shown in Table 4. And the hyperparameters of our Encoder is shown in Table 5.

Table 4: Hyparameters of our latent conditioned model $\pi_{\mathbf{z}}(\cdot|\tau_{\mathbf{z}})$.

| Hyparameter | Value |
|---|---|
| Num Layers | 3 |
| Num Heads | 2 |
| learning rate | 1.2e-4 |
| weight decay | 1e-4 |
| warmup steps | 10000 |
| Activation | relu |
| z dim | 16 |
| Value Dim | 64 |
| dropout | 0.1 |

Table 5: Hyparameters of BERT model $I_\phi$.

| Hyparameter | Value |
|---|---|
| Num Layers | 3 |
| Num Heads | 8 |
| learning rate | 1.2e-4 |
| weight decay | 1e-4 |
| warmup steps | 10000 |
| Activation | relu |
| z dim | 16 |
| Value Dim | 64 |
| dropout | 0.1 |

## B.2 Computing Resources

Our experiments are conducted on a computational cluster with multi NVIDIA-A100 GPU (40GB), and NVIDIA-V100 GPU (80GB) cards for about 20 days.

## B.3 Codebase

For more details about our approach, it can be refereed to Algorithm 1. Our source code is complished with the following projects: OPPO [1], Decision Transformer [2].

## C Supplemented Experiment Results.

In this section, we supplement all the experimental results used in Figure 4 of the main text. Table 7 corresponds to Figure 4 (a), and Table 7 corresponds to Figure 4 (b).

Table 6: Comparison of performance between GDT (multiple settings) and ContextFormer (LfD #5). Specifically, we Compare the ContextFormer and GDT with 5 expert and GDT with 5 best trajectories, ContextFormer performs the best.

| Task | Offline IL settings | GDT+5 *demo* | GDT+5 *best traj* | ContextFormer (LfD #5) |
|---|---|---|---|---|
| **hopper** | medium | 44.2± 0.9 | 55.8± 7.7 | 74.9± 9.5 |
| | medium-replay | 25.6± 4.2 | 18.3± 12.9 | 77.8± 13.3 |
| | medium-expert | 43.5± 1.3 | 89.5± 14.3 | 103.0± 2.5 |
| **walker2d** | medium | 56.9± 22.6 | 58.4± 7.3 | 80.9± 1.3 |
| | medium-replay | 19.4± 11.6 | 21.3± 12.3 | 78.6± 4.0 |
| | medium-expert | 83.4± 34.1 | 104.8± 3.4 | 102.7± 4.5 |
| **halfcheetah** | medium | 43.1± 0.1 | 42.5± 0.2 | 43.1± 0.2 |
| | medium-replay | 39.6± 0.4 | 37.0± 0.6 | 39.6± 0.4 |
| | medium-expert | 43.5± 1.3 | 86.5± 1.1 | 46.6± 3.7 |
| **sum** | | 399.2 | 514.1 | **644.9** |

Table 7: Comparison of performance between Prompt-DT, PTDT-offline and ContextFormer (LfD #1), ContextFormer performs the best. Notably, the experimental results of Prompt-DT and PTDT-offline are directly quated from Hu et al. (2023).

| Dataset | Task | Prompt-DT | PTDT-offline | ContextFormer (LfD #1) |
|---|---|---|---|---|
| medium | hopper | 68.9± 0.6 | 71.1± 1.7 | 67.8± 6.7 |
| | walker2d | 74.0± 1.4 | 74.6± 2.7 | 80.7± 1.6 |
| | halfcheetah | 42.5± 0.0 | 42.7± 0.1 | 43.0± 0.2 |
| **sum** | | 185.4 | 188.4 | **191.5** |

---

[1] https://github.com/bkkgbkjb/OPPO
[2] https://github.com/kzl/decision-transformer

