# OpenReview forum: "ContextFormer: Stitching via Latent Expert Calibration"
_TMLR — Withdrawn by Authors_

### Review · Reviewer_aKSi · 2025-06-07

**Summary Of Contributions:**

This paper proposes ContextFormer, a novel framework that enhances Decision Transformers with stitching capabilities via latent expert calibration. By framing stitching as a supervised expert matching problem, the method learns a contextual embedding z* that aligns with expert trajectories and diverges from sub-optimal ones. Unlike conservative Q-learning approaches, ContextFormer avoids OOD issues through fully supervised training in the latent space. Theoretical analysis supports its stitching mechanism, and extensive experiments show that it outperforms various Decision Transformer variants and imitation learning baselines across D4RL benchmarks.

**Audience:**

Yes

**Broader Impact Concerns:**

The paper does not include a dedicated Broader Impact Statement, and no major ethical concerns are explicitly discussed. While the proposed method, ContextFormer, is primarily evaluated in simulated environments, its potential application to real-world decision-making systems could raise concerns related to safety, robustness under distributional shift, and unintended generalization.

**Claims And Evidence:**

No

**Requested Changes:**

1. Clarify and empirically support the core observation that DT lacks stitching capability (Critical): Include targeted experiments or visualizations that isolate stitching behavior and compare DT with ContextFormer in that regard.

2. Improve clarity and rigor in the writing (Critical): Define all variables and notations clearly, and revise the structure for better readability and flow.

3. Include a more direct experimental evaluation of stitching (Important): For example, design controlled tasks or case studies that explicitly require trajectory stitching and analyze agent behavior across models.

4. Add a simple discriminative baseline (Important): Test whether training DT with expert trajectories as positives and non-expert ones as negatives (e.g., contrastive loss or binary classification-style supervision) achieves similar performance.

5. Address fairness of comparisons (Important): Discuss and ideally control for the differing treatment of expert vs. non-expert trajectories across methods, particularly in IL settings.

**Strengths And Weaknesses:**

Strengths:

1. The paper tackles an important and under-explored limitation of Decision Transformers (DT), namely their lack of stitching ability in offline reinforcement learning—a capability that is crucial for leveraging sub-optimal trajectories.

2. The proposed method, ContextFormer, is novel in its use of latent expert calibration through divergent sequential expert matching.

Weaknesses:

1. The core observation that DT lacks stitching capacity is assumed rather than empirically demonstrated or analyzed in detail.

2. Despite framing the work around stitching, the experiments do not directly measure or isolate improvements to stitching capability, nor do they include qualitative or case study-based analyses to support such claims.

3. The writing quality is poor in several places; the overall structure is difficult to follow, and many technical details are either vague or missing. For instance, key notations such as $\lambda_{1}$ in Section 5 are not clearly defined.

4. Experimental comparisons may not be entirely fair. ContextFormer treats expert and sub-optimal trajectories differently in training, whereas many baseline methods do not make this distinction, potentially giving ContextFormer an advantage.

5. The method's core idea resembles contrastive learning—pulling trajectories closer to expert demonstrations and away from non-expert ones—yet no ablation or baseline experiment is conducted to validate whether simpler discriminative setups (e.g., treating expert as positive and non-expert as negative) would suffice.

---

### Review · Reviewer_JUBX · 2025-06-09

**Summary Of Contributions:**

The paper proposes ContextFormer, a divergent sequential expert matching to utilize the sub-optimal trajectories with stitching. The key ideas involves supervised action prediction objective that avoids conservative Q-based penalties by purely matching trajectories in latent space, reducing out-of-distribution (OOD) critic issues. A theoretical analysis of stitching shows shows that minimizing the expert-matching loss directs the context vector to extract expert features from suboptimal segments, enabling effective stitching. The experiments involve encoding trajectory fragments using a BERT model for producing hindsight information vectors and training with both the supervised action prediction loss and the expert matching loss. Empirically, with as few as five expert demos, ContextFormer is compared to normalized scores by other offline IL algorithms on D4RL Mujoco tasks (Hopper, Walker2d, Ant, HalfCheetah) and various Decision Transformer variants on 2d maze tasks.

**Audience:**

Yes

**Broader Impact Concerns:**

Broader impact on improving sample efficiency for offline RL algorithms is unclear.

**Claims And Evidence:**

No

**Requested Changes:**

1. In Table 1, what is the statistical significance of these scores? Why should ContextFormer be considered the new state-of-the-art in trajectory stitching benchmarks especially among other offline IL algorithms?
1. Reference to Table 1 doesn't exist in the text yet. Reference to "table 6.6" should point to "Table 2".
1. Improve Figure 1 and 2 for clarity, high resolution and legibility
1. Clarify what is "HI" in introduction -> Hindsight Information. Perhaps elaborate what does "in-expert distribution HI from sub-optimal datasets" mean?
1. typos: ~detials~ -> details, ~vice visa~ -> vice versa
1. Broader impact section should be present

**Strengths And Weaknesses:**

### Strengths
- ContextFormer replaces conservative Q-value penalization with a clean supervised latent matching objective, reducing OOD brittleness.
- The paper provides theoretical links between the loss and the selection of “in-expert” fragments.

### Weaknesses
- Experiments are limited to simulated Mujoco and maze2d tasks; real-world or higher-dimensional domains remain untested.
- Adds overhead of updating first, the HI extractor, then the contextual latent vector z*, and finally the policy.
- ContextFormer focuses on purely offline settings; how it integrates with online fine-tuning or continual learning is left open.

---

### Review · Reviewer_rssJ · 2025-06-09

**Summary Of Contributions:**

This work introduces ContextFormer, which enhances the decision transformer with a latent representation learning that enhances the "stitching" capability of DTs. "Stitching" refers to the ability of offline RL methods to stitch together suboptimal trajectory fragments to create better ones in order to better utilize offline data. ContextFormer learns latent representation from the expert and suboptimal trajectories using a BERT-like encoder, and conditions the decision transformer on the learned latent representation to stich expert behaviors. Experiments on the D4RL benchmark show that ContextFormer offers significant performance gains against DT variants.

**Audience:**

Yes

**Broader Impact Concerns:**

I do not see any ethical concers of this work, as it mostly deals with simulated offline RL/IL benchmarks.

**Claims And Evidence:**

Yes

**Requested Changes:**

- Since sub-optimal data stiching is the key innovation, providing samples showing how noisy expert data can affect the method will help strengthen the claims of this work.

- Additional discussion on computational overhead and cost would be beneficial.

- Discussion on how to scale ContextFormer to more complex tasks (higher dimensional control, manipulation, visual-control, etc.) will be helpful in putting this work into more context.

- Intuitive examples of stitching would be beneficial to include concrete, intuitive examples or visualizations (e.g., showing how trajectory fragments from different demonstrations are merged via z*) to help readers better understand how the model performs stitching in practice..



Minor Issues:
page 2: "aren't suffered from"
page 5: "the the"

**Strengths And Weaknesses:**

**Strength:**

- This work addresses a known limitation of DT, which is essential for better leveraging offline RL datasets where additional interactions is not possible.
- The proposed supervised latent matching procedure avoids conservatism from Q-learning approaches like QDT that suffer from out-of-distribution (OOD) and over-penalization.
- The proposed divergent sequential expert matching method is theoretically grounded (though I am not an expert) and sound.
- Strong empirical results show that ContextFormer outperforms prior IL and DT variants on D4RL and Maze2D datasets under controlled settings.



**Weakness:**
- Experiments are restricted to Gym-Mujoco and Maze2D. In this day and age of real-world behavior cloning and robotics, the method’s generalizability to high-dimensional tasks (e.g., image-based, sparse reward, or multi-agent environments) is not assessed.
- The use of BERT-style encoders and multiple optimization loops introduces additional complexity and compute overhead, but the trade-offs vs. simpler baselines (for instance, DT with better prompt tuning) are not compared.

---

### Note · Authors · 2025-07-27

**Comment:**

Dear Action Editor and Reviewers,

I hope this email finds you well! I am writing to inform you that, unfortunately, due to ongoing health issues and after receiving medical treatment, I am still not showing significant improvement. As a result, I find myself unable to continue with the necessary work to address the comments.

After careful consideration, I regretfully request to withdraw my manuscript from the review process. I deeply apologize for any inconvenience this may cause to you, the reviewers, and the editorial team. Please understand that this decision comes under circumstances beyond my control, and I sincerely appreciate your understanding and support during this challenging time.

I would like to express my heartfelt gratitude to the AE and reviewers for their time and effort in reviewing my work. I regret that I am unable to fully address their valuable feedback, but I hope that I will have the opportunity to submit my work for consideration again once my health permits.

Thank you for your time, understanding, and continued support. I look forward to hearing from you regarding the withdrawal process.

Sincerely,
Authors

**Withdrawal Confirmation:**

I have read and agree with the venue's withdrawal policy on behalf of myself and my co-authors.